# Gradual Provision of Live Black Soldier Fly (*Hermetia illucens*) Larvae to Older Laying Hens: Effect on Production Performance, Egg Quality, Feather Condition and Behavior

**DOI:** 10.3390/ani10020216

**Published:** 2020-01-28

**Authors:** Laura Star, Tarique Arsiwalla, Francesc Molist, Raymond Leushuis, Monika Dalim, Aman Paul

**Affiliations:** 1R&D, Schothorst Feed Research B.V., 8218 NA Lelystad, The Netherlands; LStar@schothorst.nl (L.S.); Fmolist@schothorst.nl (F.M.); 2Product Development, Protix B.V., 5107 NC Dongen, The Netherlands; TariqueArsiwalla@hotmail.com (T.A.); Raymond.Leushuis@protix.eu (R.L.); Monika.Dalim@protix.eu (M.D.)

**Keywords:** *Hermetia illucens*, live feeding, older laying hens, production performance, egg quality, feather condition

## Abstract

**Simple Summary:**

In nature, hens spend considerable amounts of time eating live insects. This is considered as their natural behavior and may positively contribute to animal welfare. However, laying hens generally have limited access to insects in current intensive farming systems. *Hermetia illucens* larvae are nutritious and can be industrially produced using the principles of circular agriculture. In Europe, legislation allows the feeding of live insects to poultry, and could possibly be used to replace soy in diets of laying hens as protein source. The majority of soy meal used in Europe originates from North and South American countries. Increasing soy plantations in South American countries is often linked to deforestation and social issues. This research evaluated effects of including live *H. illucens* larvae, as replacement of soy in the daily ration, on production performance, egg quality, behavior and feather condition of older laying hens. Live *H. illucens* larvae can be used in combination with local plant proteins to successfully replace soy in diets of older laying hens. Feeding hens live *H. illucens* larvae also had a positive effect on the feather condition of birds.

**Abstract:**

Feather pecking is a key welfare challenge in laying hen husbandry. Feeding of live *Hermetia illucens* larvae could provide a possible solution to reduce feather pecking in hens. This research investigates effects of dispensing live *H. illucens* larvae to non-beak trimmed older laying hens on production performance, behavior and welfare. Control treatment hens were provided a commercial diet, while larvae treatment hens were provided live *H. illucens* larvae (using special dispenser) on top of a soy-free diet. Feather condition, production performance and egg quality were measured during the initiation (67 weeks age) and termination (78 weeks age) of the trial. Behavior of birds was monitored using video recording. Feed conversion ratio, body weight gain and egg laying parameters were similar for both treatments. At termination of the trial, larvae-fed hens exhibited better feather condition in comparison to control hens (*p* = 0.004). Behavioral observations indicated that larvae provision influenced the number of birds on floor during morning and afternoon hours. In conclusion, live *H. illucens* larvae could successfully replace soy in diets of older laying hens (in combination with local plant proteins). Provisioning of these insects also had a positive effect on the feather condition of laying hens with intact beaks.

## 1. Introduction

Ancestors of modern poultry lived in social groups of 20 to 30 chickens. In current non-cage systems, the flock sizes are much larger. Laying hens are not able to remember or recognize all flock mates in such systems and there is complete absence of social hierarchy [1]. This results in feather pecking, which is one of the greatest challenges in commercial laying hen husbandry [2]. At present, non-cage systems are prevalent in Europe and North America. Furthermore, in future South American, Asian and African countries are also expected to adopt non-cage poultry systems [3]. Therefore, impact of severe feather pecking will further increase.

Laying hens learn to peck at young age [4]. Severe feather pecking is often evident in old layer hens [4,5]. Provisioning of pecking substrate at young age may reduce the likelihood of sever feather pecking at maturity [6]. However, such provision does not completely eliminate the risk of severe feather pecking during adulthood [7]. Some studies have indicated that pecking is more severe in old laying hens (>70 weeks of age) when compared to young hens (25 weeks of age) [8,9]. This makes ensuring the welfare of old laying hens more challenging. Some authors have also indicated that inability of chicken to express their natural behavior can result in aggressive behavior and increased pecking [10]. Presentation of live invertebrate to consuming animals (natural carnivores or omnivores), provides them an opportunity to express their natural behavior [11]. Expression of natural behavior is likely to be pleasurable for the animal [12]. Chickens are effective foragers of live insects [13], since insects are part of the natural diet of chickens. Free-range chickens spend about 37% of their time looking for and eating insects [14]. Insects not only present a moving stimuli to attract the attention of chickens, but are also nutritious [15,16,17,18].

Live insects, including black soldier fly larvae (*Hermetia illucens*) are already approved for poultry feeding in Europe [19]. Black soldier fly larvae have gained special status amongst insects due to their unique: (a) ability to consume a wide range of organic side streams; and (b) nutritional profile, especially the protein composition [20].

A recently published report indicated that inclusion of dried black soldier fly pre-pupae meal improved the egg weight and egg-shell thickness in comparison to a control diet (no pre-pupae meal inclusion). However, no differences in production performance were observed [21]. A review article published in 2019 compiles literature related to inclusion of black soldier fly larvae meal in poultry feed [22]. However, to the best of the authors’ knowledge, there is no peer reviewed literature studying the inclusion of live black soldier fly larvae in laying diets. It is expected that provision of live black soldier fly larvae will not adversely affect the production performance and egg quality parameters. Another topic that remains to be investigated is the provisioning of live black soldier fly larvae in relation to the welfare of older laying hens (where welfare is more challenging than younger birds). Current research was conducted to evaluate the effect of steady and random provisioning of live black soldier fly larvae to older laying hens on: (a) production performance; (b) egg quality; (c) feather condition; and (d) bird behavior.

## 2. Materials and Methods

### 2.1. Laying Hens and Housing Management

Older laying hens (Dekalb White; 65 weeks of age) arrived at Schothorst Feed Research (SFR; Lelystad, The Netherlands) in the last week of December 2018. These hens were allocated to aviary pens with wood shavings on the floor and allowed to adapt (acclimatize) for a period of 2 weeks. Hens were 67 weeks of age during the initiation of the trial. Before arriving at the experimental facilities hens were housed in an aviary system (Vencomatic, Eersel, The Netherlands) with 330 birds per pen.

Only hens that were active and showed no clinical signs at 67 weeks age were included in the trial. Hens were not identified by a separate individual code, rather they were grouped and identified using unique pen number. After initiation of the trial, hens were excluded if they got sick or met humane end-points. Humane end-points are defined as situations in which laying hens were clinically sick without prospects of recovery, severely injured, or when hens were unable to stand upright. Beak treatment of laying hens (to avoid damage by severe pecking) is not allowed in The Netherlands, so all birds in this trial had intact beaks.

Feeding trials were realized in two identical houses that were windowless, artificially lighted, and centrally heated (target temperature: 20 ± 2 °C). Laying hens were accommodated in aviary pens of 1.5 m length (including laying nest), 2 m wide and 2.3 m in height. Hens were accommodated at a density of 22 hens/pen. Pens were equipped with perches (approx. 18 cm/hen), a feeder bin (ad libitum feed, approx. 5 cm/hen feeder space) and six nipple drinkers per pen (ad libitum water). Bedding consisted of fresh wood shavings.

Pens were inspected every day during the trials for specific observations (e.g., health of the birds) and recorded. The hens were not vaccinated during the trial.

### 2.2. Black Soldier Fly Larvae and Larvae Dispenser

Live black soldier fly larvae were supplied by Protix B.V. (Dongen, The Netherlands). Larvae were produced in GMP+ and SecureFeed certified facility under HACCP (Hazard Analysis Critical Control Points) conditions. Fresh and live larvae were supplied on a weekly basis, and were stored in a cool and dry place until consumed. The nutritional composition of the live larvae as declared by the manufacturer is indicated in Table 1.

The larvae dispenser used during the study was also supplied by Protix B.V. (Dongen, The Netherlands). Design of the dispenser is indicated in Figure 1. The larvae dispenser was designed such that approximately 275 g of live larvae could be dispensed from the four exits of the dispenser (equally and randomly) during a 6 h period. The dispenser consisted of a buffer that stored required amounts of larvae for one day of feeding. Larvae gradually fell from the buffer into the dispenser unit which further bifurcated the larvae into one of the four discharge points. The size and volume of the dispenser was designed keeping in mind the number of hens (so that every chicken got the same proportion of larvae) and dimensions of the pen.

### 2.3. Experimental Diets

Commercial laying mash diet (containing soy) and a special soy-free diet were supplied by ABZ Diervoeding (Leusden, The Netherlands). The ingredients and calculated composition of diets are presented in Table 2 and Table 3, respectively. Both diets were formulated to meet the nutritional requirements of older laying hens.

### 2.4. Study Design and Feeding Regime

A randomized complete block design was used for the experiment with two treatments (control and larvae-fed) and eight replicates (22 hens/replicate at the start of the trial). Blocking was applied to the position of pens in the experimental facility. The trial was conducted for a duration of 12 weeks.

Control groups (Group A) were provided with soy containing commercial laying mash diet. The larvae-fed group (Group B) were provided with a soy-free diet. Both groups were provided ad libitum feed and water. On the top of soy-free diet, Group B hens were also provided with 12 g live larvae per hen per day (10% of daily feed intake) using larvae dispenser described in Section 2.2. Larvae were provided at 11.30 am each day, and the dispenser (Figure 1) self-emptied at approximately 5.30 p.m.

### 2.5. Production Performance and Mortality

The following production parameters were measured during the study: (1) body weight gain from initiation until the end of the feeding trial; (2) weekly feed intake/pen; (3) weekly number of eggs/pen and egg weight/pen; (4) laying rate, egg mass and feed conversion ratio were calculated from the above data. Mortality was recorded daily. Hens that were sick, severely wounded, or could not stand upright were euthanized.

### 2.6. Egg Quality

Quality parameters, i.e., egg-shell breaking strength, elasticity of shell and Haugh unit were evaluated for ten eggs/pen during initiation and termination of the trial. These parameters were evaluated by Institute of Quality Measurement in Eggs (Amersfoort, The Netherlands).

### 2.7. Feather Condition

Laying hens (5 birds/pen) were randomly chosen and evaluated for feather condition during initiation and termination of the feeding trial. Feather condition was scored between 0 (intact feathers with no injuries or scratches) to 5 (completely denuded area). Neck, back, rump and belly were taken into account for scoring, because of their association with feather pecking behavior [23,24].

### 2.8. Bird Behavior

Video observations were made in 2 pens (1 from Group A and 1 from Group B) to record and analyse hen behavior. The scheme used for video observation is mentioned in Table 4. Video recordings were made to have undisturbed observations. Behavior was scored by counting the number of birds on the floor at an interval of every 5 min during the observation period. Video observations were recorded to understand the influence of larvae provisioning on birds’ behavior.

### 2.9. Data Exclusion Parameters and Statistical Analysis

A specific observation was marked outlier and excluded from the dataset before statistical analysis, if the residual (fitted—observed value) was greater than 2.5 times standard error of the residuals of the data set (ANOVA). If a specific observation on feed intake, laying rate, or egg weight was considered outlier, it was not used for the calculation of corresponding feed conversion ratio and egg mass (were excluded).

Experimental data were analysed using GenStat^®^ version 19.1 (VSN International Ltd., Hemel Hempstead, UK). Feed intake, laying rate, egg mass and weight, feed conversion ratio, mortality, egg quality parameters and feather scores were compared between two treatments using ANOVA. Pen was the experiment unit. General model was:Y_ij_ = µ + Block_i_ + Treatment_j_ + e_ij_(1)

With:

Y_ij_ = response parameter; µ = overall mean; Block_i_ = block effect (i = 1 to 8); Treatment_j_ = effect of treament group (j = 1, 2); e_ij_ = residual error; Values with *p* ≤ 0.05 were considered statistically different.

### 2.10. Animal Ethics

The trial was realized according to the guidelines of the Animal and Human Welfare Codes/Laboratory practice codes in The Netherlands. Trial protocol was approved by the Schothorst Feed Research Institute Ethics Review Committee.

## 3. Results

### 3.1. Production Performance

Results corresponding to feed intake, laying rate, egg weight, egg mass and mortality rate are presented in Table 5 and Table 6. Group B (larvae-fed hens) had a significantly lower (*p* = 0.029) feed intake in comparison to Group A (control). Feed intake indicated in Table 5 is based on intake of mash diets (larvae intake by Group B not taken into account). Group B hens received approximately 12 g of larvae per hen per day (g/h/d). Therefore total feed intake for Group B was about 135 g/h/d, which was not different from Group A. Larvae consist of approximately 70% moisture, so on dry matter basis, feed intake of Group B was 127 g/h/d, which is numerically lower than the control treatment (significant differences were not calculated). Laying rate, egg weight, egg mass and mortality rate did not differ between treatments.

Feed conversion ratios of both treatments are also indicated in Table 5. In line with a lower feed intake, Group B also showed a significantly lower feed conversion ratio (*p* = 0.004). If larvae intake by Group B hens is taken into account on a dry matter basis, feed conversion ratio of Group B is estimated to be 2.452. This value is numerically lower than the feed conversion ratio of Group A and is only significant at the 10% level of confidence (*p* = 0.071).

Body weight of laying hens was determined during the initiation (67 weeks of age) and termination (78 weeks of age) of the trial. Body weight of hens during the trial is indicated in Table 7. Group A hens showed an average weight loss during the trial, while Group B hens showed an average weight increase. However, there was no significant difference between the two experimental groups.

### 3.2. Egg Quality

Egg quality parameters were determined during the initiation and termination of the trial. Values obtained for quality parameters are mentioned in Table 8. There were no differences between egg-shell strength, elasticity and Haugh unit between eggs from Groups A and B hens.

### 3.3. Feather Condition

The feather condition scores of the laying hens determined during the initiation and termination of the trial are indicated in Table 9. Older laying hens already had feather damage during the initiation of the trial. Initial feather damage of Group B was numerically higher than Group A (no significant differences, *p* = 0.06). At the end of the trial, feather damage of Group B was significantly less compared to Group A hens (*p* = 0.004). Some hens in both groups started to molt during the trial, which is marked by renewing of feathers. During the initiation, all the scored hens had a bald belly (corresponding to score 4 or 5). However, during termination five laying hens were scored ≤ 2 for belly. If these molted hens are taken into account, feather condition score for Groups A and B could be adjusted to 3.3 and 2.6, respectively. Even after adjustment of the score, feather damage of hens from Group B was significantly less compared to hens from Group A (*p* < 0.05).

### 3.4. Bird Behavior

Video observations of bird behavior were made during the initiation and termination of the trial. Analytical data revealed higher counts of hens on floor during morning hours in Group B (when larvae were loaded in dispenser) when compared to Group A (Figure 2). Whereas, for Group A higher counts of hens on floor were observed during afternoon hours when compared to Group B.

## 4. Discussion

### 4.1. Effect on Production Performance

Groups A and B hens had a feed intake of 133 and 123 g/h/d, respectively (without accounting larvae intake by Group B). The reduction in mash feed is linked to the nutritional quality of live larvae, which are able to complete the proportion of protein and fat in diets (Table 6). Live larvae used during the current study originated from GMP+ and Securefeed certified factory and were grown using HACCP principles, indicating high nutritional quality.

After adjusting the larvae intake by laying hens, the feed conversion ratio of larvae-fed hens was numerically lower than control hens. Even though the differences were not significant, but *p*-value (0.071) approached the borderline of significance.

There were no differences in the laying rate and egg weight between Groups A and B. It appears that live larvae inclusion in soy-free diets made from local ingredients (in this case rapeseed meal) had no adverse effect on egg production. In Europe, soy is the major source of protein used in poultry diet formulations [25]. Approximately 32 g soy meal is consumed by hens to lay every single egg [26]. The majority of soy being used for this purpose originates from American countries. According to some estimates, 79% of soy meal consumed in EU originates from South America [27] with Brazil alone supplying more than 40% of the total consumption [28]. Import of soy from South American countries has been a subject of debate in recent years. Increasing soy plantations in these countries has been frequently associated with deforestation, which further translates into increasing threat to indigenous people and human right violations [29]. Major drivers of soy-trade linked deforestation in South America are: (1) increasing profit per hectare from soy plantation; (2) development of infrastructure for e.g., roads to facilitate soy transport; and (3) badly fabricated regulations [26]. This scenario indicates the urgent need to find soy meal substitute. Insects could be grown using wide range of agro-food industry by-products, serving as an important pillar of local circular economy [20]. From the outcomes of this study it is suggested that live *Hermetia illucens* larvae (in addition to other plant protein sources) could be successfully used for the replacement of soy in European poultry diets without detrimental effects on production performance, behavior and welfare of older layer hens.

A slight average increase in body weight was observed for Group B hens (contrary to Group A hens). However, there were no significant differences in body weight between both groups. For laying hens at this age, body weight gain has very little significance. Body weight gain is more significant for broiler industry, which could be a subject of future work.

### 4.2. Effect on Egg Quality

Egg quality parameters (i.e., shell strength, eleasticity and Haugh unit) were unchanged with or without inclusion of larvae in diets. This finding adds to the body of evidence that black soldier fly larvae together with a local plant protein source can replace soy in poultry diets.

A previous research already investigated the impact of black soldier fly larvae protein meal based diets on quality of eggs produced by laying hens [30]. At 5% inclusion levels, better egg-shell strength was observed in comparison to zero inclusion levels. In the current study, 12 g larvae together with 123.3 g mash feed, corresponds to 1.3% black soldier fly larvae protein inclusion. Therefore, in the future it could be interesting to investigate the effect of increased larvae inclusion rates on egg quality. Studies have also indicated the beneficial effect of black soldier fly protein meal inclusion on yolk color, γ-tocopherol, lutein, β-carotene and cholesterol content [30,31]. Inclusion of live larvae on the concentration of these molecules could be another subject of future research.

### 4.3. Effect on Feather Condition

During the initiation of the trial, feather damage of larvae-fed hens was numerically higher than control hens (no significant differences). However, the *p*-value was 0.06, indicating difference near significant trend. Outcomes of current research indicate that provision of live black soldier fly larvae in a random and controlled manner resulted in reduction of feather damage in older laying hens. Even though larvae-fed hens started with bad feather score (in comparison to control hens), these hens ended up having significantly better feather score during the termination of trials.

Gentle pecking is normal in laying hens and results in little or no feather damage [32]. Feather pecking is affected by: (a) internal factors: genetic strain, age, hormonal state, fearfulness and social motivations; and (b) external factors: floor substrate, flock size/density, light intensity and diets [33]. Amongst these factors, relation between age and feather pecking is particularly important for egg producers and consumers. Laying hens are commonly used for egg production until 86 weeks [34,35] and in The Netherlands it is even increasingly common to keep white laying hens until 95 to 100 weeks of age (without molting). In laying hens of age above 65 weeks, severe feather pecking is extremely common [8,33]. Some widely used commercial solutions to restrict pecking involve either trimming of the beaks or keep hens in small and confined groups. However, these techniques have their own welfare issues [36]. The first solution is not allowed in The Netherlands, where the study was performed. Therefore, all laying hens used had intact beaks, contributing to the poor feather condition of the older laying hen at the start of the trial. Other pathways to reduce feather pecking include: offering enrichments (natural substrate to peck at), lower stocking density (facilitating hens to remember flockmates), and adapted feed formulation (increasing proteins levels in diet) [37]. However, all these systems still include usage of soy-based diets.

Alternatives such as free-range systems are also considered effective in reducing feather pecking. A study indicated that free-range chickens spend about one-third of their time eating insects, which provides them with a natural substrate to peck at [14]. However, recent findings have indicated that high mortality in a free-range system is negative in relation to animal welfare [38]. Normally, laying hens in closed systems have little access to live insects (besides pest insects). Providing a sufficient amount of larvae (through a specially designed larvae dispenser) to each hen offered a natural substrate to peck at and distract the laying hens from pecking at each other. Similar effects were also observed in young turkey poults. Feeding live black soldier fly larvae resulted in reduced feather pecking on back and tail base [39]. It could also be of future interest to study the effect of providing live black soldier fly larvae to laying pullets on development of pecking behavior.

### 4.4. Effect on Bird Behavior

Live larvae-fed hens (Group B) counts on the floor were higher during the morning than the afternoon. Group A hens (control) counts on the floor were lower in the morning compared to Group B hens. However, Group A hens counts on the floor were higher in the afternoon compared to Group B hens. It might be suggested that the supply of larvae in the morning fulfilled the need of laying hens to show feed searching behavior (i.e., their behavior was rewarded). Results from the current research indicate that provision of live black soldier fly larvae could facilitate hens in expressing their natural behavior. Expression of natural behavior is also linked to reduced feather pecking in case of laying hens [40] and will contribute to bird welfare.

Researchers have indicated that higher pecking activity has been observed in laying hens during morning hours [41]. These birds spend a more time being quiet, sitting and resting during the afternoon [42]. Video observations for larvae-fed hens also indicated higher bird counts on the floor during morning hours. Besides, these birds had a better feather condition at the end of the trial. It could be hypothesized that larvae-fed birds spend more time pecking at larvae than at each other. Additionally, larvae availability in the morning was satisfying, providing them an opportunity to rest during the afternoon. Lastly, in the future, it could also be of interest to optimize the live larvae provision time to sufficiently satisfy the hens. Live insects, particularly black soldier fly larvae, are already approved for poultry feeding in Europe. Several companies are now engaged in the business of rearing black soldier fly larvae using food industry by-products. These insects are now being viewed as important pillars of circular economy [20]. As a result, there is a substantial interest in feeding black soldier fly larvae to laying hens and broilers [30,31,39,43]. This study provides an interesting first insight regarding the successful inclusion of live black soldier fly larvae in diets of older laying hens. However, this study is potentially incomplete without evaluation of (a) performance and feather pecking development in young laying hens; (b) nutritional egg quality; (c) other welfare traits; and (d) additional costs involved in relation to the benefits. Furthermore, this study could also be expanded to broilers, breeder flocks and turkeys.

## 5. Conclusions

There is a growing interest in utilization of live black soldier fly larvae for feeding poultry. Literature indicates that these insects are nutritious and their farming could facilitate circular economy [20]. Moreover, these insects could also help in solving some welfare issues related to poultry farming, e.g., feather pecking [43]. Keeping this in mind, the current study was realized to study the effect of: (a) feeding soy-free diets containing combination of local plant proteins and black soldier fly larvae to older laying hens on production performance and egg quality; and (b) effect of dispensing live black soldier larvae to older laying hens on feather condition and hen behavior.

Results obtained during the study indicated that replacing soy with live black soldier fly larvae and local protein sources has no adverse effect on production performance and egg quality. Additionally, random and steady provision of larvae to older laying hens with intact beaks had a positive effect on feather condition.

## Figures and Tables

**Figure 1 animals-10-00216-f001:**
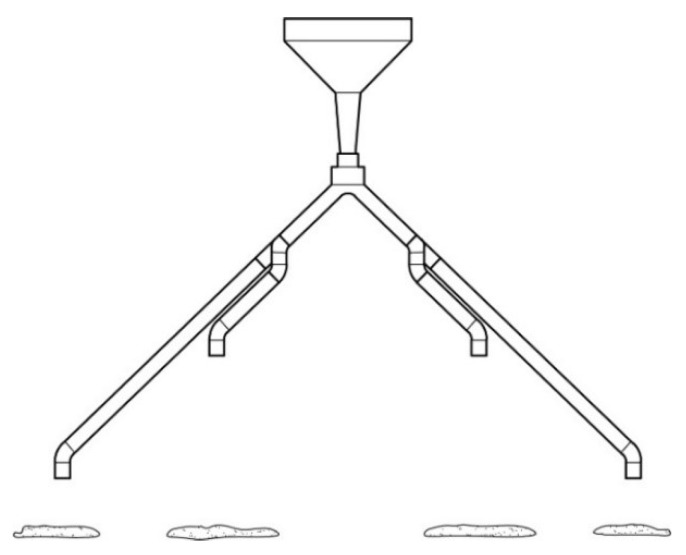
Live black soldier fly larvae dispenser.

**Figure 2 animals-10-00216-f002:**
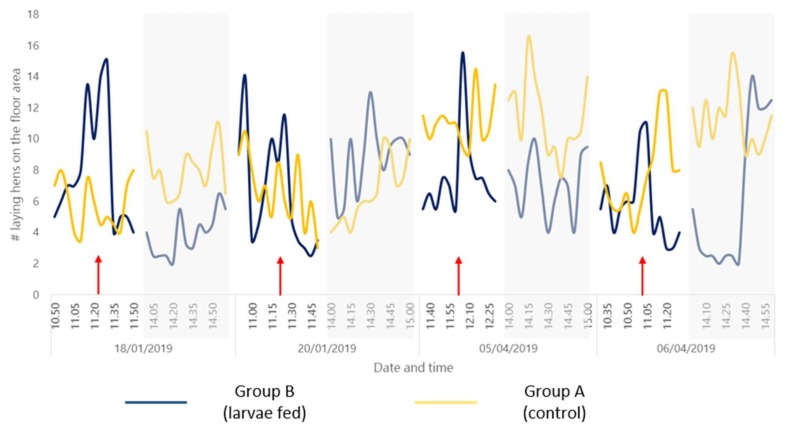
Number of hens observed at the floor area of pen in Group A (control) and Group B (larvae fed) at the initiation (18/01/2019 and 20/01/2019) and termination (05/04/2019 and 06/04/2019). Observations were performed in morning (after loading of larvae in dispenser, approx. 11.00 a.m. to 12.00 p.m.) and afternoon (2.00 p.m. to 3.00 p.m.). Red arrows indicate time of larvae supply.

**Table 1 animals-10-00216-t001:** Nutritional composition of live larvae (as in basis, provided by supplier).

Nutrients	Live Larvae
Moisture (g/kg)	700.0
Crude protein (g/kg)	135.0
Crude fat (g/kg)	105.0

**Table 2 animals-10-00216-t002:** Ingredient composition of experimental diets.

% Ingredients	Commercial Mash Diet	Soy-Free Mash Diet
Maize	30.00	30.00
Wheat	34.61	35.52
Soybean meal > 48% CP	9.70	0.00
Sunflower seed meal 38% CP < 1% CF	6.00	6.00
Rapeseed meal (SE)	2.50	7.42
Poultry fat	2.36	2.07
Maize gluten meal > 60% CP	2.22	5.16
Alfalfa 16–19% CP	1.02	0.00
Potato protein	0.00	1.89
Limestone	8.88	8.90
Monocalcium phosphate	0.23	0.23
Sodium bicarbonate	0.25	0.27
Potassium bicarbonate	0.03	0.32
Salt	0.02	0.00
Premix ^1^	1.00	1.00
Lysine-HCl (L 79%)	0.13	0.21
Methionine (DL 99%)	0.03	0.00
Premix red	0.50	0.50
Phytase	0.26	0.27
NSP-enzyme	0.25	0.25

^1^ Containing: Cu (CuSO_4_.5H_2_O) 1,000 mg/kg; Fe (FeSO_4_. H_2_O) 4000 mg/kg; Mn (MnO) 10,000 mg/kg; Zn (ZnSO_4_.H_2_O) 4400 mg/kg; I (Ca(IO_3_)_2_ anhydrous) 100 mg/kg; Se (Na_2_SeO_3_) 15 mg/kg; vit. A 750,000 IU/kg, vit. D_3_ 150,000 IU/kg; vit. E 1250 IU/kg; pantothenic acid 500 mg/kg; niacin 1000 mg/kg; vit. B6 100 mg/kg; vit. B12 2000 μg/kg; biotin 4000 μg/kg; vit. K3 200 mg/kg; choline 20,000 mg/kg; DL-methionine 100 g/kg.

**Table 3 animals-10-00216-t003:** Calculated nutrient composition of experimental diets (as in basis).

Nutrients	Commercial Mash Diet	Soy-Free Mash Diet
Energy (kcal/kg)	2800.00	2800.00
Moisture (g/kg)	112.00	111.00
Ash (g/kg)	124.00	121.00
Crude protein (g/kg)	158.00	160.00
Crude fat (g/kg)	50.00	48.30
Crude fibre (g/kg)	32.00	32.70
Starch (g/kg)	387.00	396.00
Ca (g/kg)	38.00	38.00
P (g/kg)	3.93	4.01
Na (g/kg)	1.50	1.50
Cl (g/kg)	1.80	1.80
K (g/kg)	6.47	6.48
Lysine (g/kg) ^1^	6.40	6.40
Methionine + cysteine (g/kg) ^1^	6.08	6.34
Threonine (g/kg) ^1^	4.54	4.80
Tryptophan (g/kg) ^1^	1.54	1.41

^1^ Based on apparent fecal digestibility.

**Table 4 animals-10-00216-t004:** Video observation regime.

Period	Corresponding Observation Day	Morning	Afternoon
Beginning of trial	First day (18/01/2019)	11.00 a.m. to 12.00 p.m. ^1^	2:00 p.m. to 3:00 p.m. ^2^
Beginning of trial	Second day (20/01/2019)		
Termination of trial	Second last day (05/04/2019)		
Termination of trial	Last day (06/04/2019)		

^1^ 30 min before and after loading of dispenser; ^2^ No interruption time.

**Table 5 animals-10-00216-t005:** Production performance and mortality rate of laying hens fed a commercial diet (Group A) or a soy-free diet + live larvae (Group B) from 67 to 78 weeks of age.

Treatment	Feed Intake ^2^ (g/h/d)	Laying Rate (%)	Egg Weight (g)	Egg Mass (g/d)	Mortality (%)	Feed Conversion Ratio (g/g)
Group A	133 ^a^	83.3	63.11	52.58	2.8	2.534
Group B	123 ^b^	81.9	63.32	51.79	1.1	2.391
SEM ^1^	2.538	1.893	0.153	1.193	0.845	0.0238
*p*-value	0.029	0.601	0.353	0.657	0.197	0.004

^a,b^ Values without a common superscript in a column differ significantly (*p* ≤ 0.05). ^1^ SEM = Standard error of means. ^2^ Intake of larvae by Group B was not taken into account.

**Table 6 animals-10-00216-t006:** Total crude protein and fat intake by laying hens fed with a commercial diet (Group A) or a soy-free diet + live larvae (Group B) from 67 to 78 weeks of age (as in basis).

Parameters	Group A	Group B
Nutrient composition of diets ^1^		
Crude protein (g/kg)	158.0	160.0
Crude fat (g/kg)	50.0	48.3
Energy (kcal/kg)	2800.0	2800.0
Nutrient composition of larvae ^2^		
Crude protein (g/kg)	-	135.0
Crude fat (g/kg)	-	105.0
Total feed and nutrient intake		
Feed intake (g/h/d)	133.1	123.3
Larvae intake (g/h/d)	0	12.0
Crude protein intake (g/d)	21.0	21.3
Crude fat intake (g/d)	6.66	7.21

^1^ Value provided by the supplier of mash feed and live larvae. ^2^ Nutrient composition of the larvae based on provided information of larvae supplier.

**Table 7 animals-10-00216-t007:** Body weight (g) of laying hens fed a commercial diet (Group A) or a soy-free diet + live larvae (Group B) from 67 to 78 weeks of age.

Treatment	67 Weeks (g)	78 Weeks (g)
Group A	1669	1660
Group B	1664	1675
SEM ^1^	11.1	16.2
*p*-value	0.752	0.529

^1^ SEM = Standard error of means.

**Table 8 animals-10-00216-t008:** Quality parameters of eggs from 67 and 78 weeks age laying hens fed a commercial diet (Group A) or a soy-free diet + live larvae (Group B).

Treatment	Egg Weight ^2^ (g)	Breaking Strenght (N)			Elasticity (N/S)		Haugh Unit	
	67 Weeks	78 Weeks	67 Weeks	78 Weeks	67 Weeks	78 Weeks	67 Weeks	78 Weeks
Group A	63.54	62.86	37.7	38.9	536	623	79.3	75.6
Group B	63.80	63.07	39.5	38.7	541	595	79.5	77.0
SEM ^1^	0.607	0.825	0.871	1.038	9.20	19.43	0.776	1.875
*p*-value	0.768	0.856	0.181	0.886	0.710	0.347	0.836	0.619

^1^ SEM = Standard error of means. ^2^ Egg weight determined only for eggs used for quality measurement.

**Table 9 animals-10-00216-t009:** Feather condition score of laying hens fed a commercial diet (Group A) or a soy-free diet + live larvae (Group B) from 67 to 78 weeks of age.

Treatment	Feather Score ^2^	
	67 Weeks (g)	78 Weeks (g)
Group A	3.4	2.9 ^a^
Group B	3.6	2.2 ^b^
SEM ^1^	0.077	0.107
*p*-value	0.060	0.004

^a,b^ Values without a common superscript in a column differ significantly (*p* ≤ 0.05). ^1^ SEM = Standard error of means. ^2^ Feather condition score from 0 (intact feathers, no injuries or scratches) to 5 (completely denuded area) were scored for neck, back, rump and belly per hen. Average feather condition score was calculated and analysed.

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
