# Peer review of "Gradual Provision of Live Black Soldier Fly (Hermetia illucens) Larvae to Older Laying Hens: Effect on Production Performance, Egg Quality, Feather Condition and Behavior"

_animals, 2020, doi:10.3390/ani10020216_

Round 1
Reviewer 1 Report
Dear authors,
Thank you for the revision of your manuscript.
The topic of feeding insects to poultry (and its place in circular economy) is on the doorstep of big changes. This manuscript may contribute to this great change!
Please find included some minor revisions.

Author Response
Dear Reviewer,
Thanks for investing your time in improving the quality of the manuscript. All your changes have been included in the updated version (attached). Here is the reply to your comment:
Comment 1: This is a very nice addition to the M&M. Very much appreciated.
Reply- Thanks for your comment.
Hope the manuscript is suitable for publication now.
Kind regards,
Aman Paul

Reviewer 2 Report
The authors have satisfactorily addressed the comments provided in the review and it is the opinion of this referee that the manuscript is suitable for publication
Author Response
Dear Reviewer,
Thanks for the comforting words.
Kind regards,
Aman Paul
This manuscript is a resubmission of an earlier submission. The following is a list of the peer review reports and author responses from that submission.
Round 1
Reviewer 1 Report
With great interest did I read and review the manuscript!
The topic is relevant, ongoing and has a lot of potential for future developments. The researcers are encouraged to further investigate the topic of feeding insects (either live or as meal) to poultry and other livestock species.
The manuscript does need some modifications. These are suggested in the attachement.

Author Response
Dear Reviewer 1,
We really believe that your comments positive improved the quality of manuscript. We have addressed most of your comments in the revised manuscript (attached). Your comments are visible in red font with no shading (in all markup version of word file). Find below the reply to your individual comments:
X1- Recommended term is ‘laying hens’. Please change throughout the whole manuscript
AP- Ok
X2- Feather pecking is a problem in young and old laying hens. If learned at young age, the hens will continue to express the damaging behavior throughout their life. FP is also no only an issue or of extra interest in countries where beak trimming will be prohibited. It was a problem before the ban on beak trimming as well.
Furthermore, worldwide more hens will be kept under cage-free conditions (e.g. in Latin America and Asia). In these regions beak trimming is still allowed, but FP will be an emerging issue. Please verify this information in the introduction with recent literature to emphasize the relevance of FP.
AP- We agree with reviewers comments.
Laying hens learn FP at young age- Included in introduction section (Line 50) Beak trimming- removed from text Cage free system in future- Included in introduction section (Line 48)
X3- (p = ….)
AP- P value is now included in abstract.
X4- Results of bird behaviour (time spent on the floor in morning/afternoon) is missing
AP- Results now included in abstract.
X5- Repetition of previous sentence
AP- We disagree with the reviewer. First statement is on replacement of soy and second statement is on feather pecking.
X6- Of what?
X7- Of what?
X8- Is this the correct terminology?
X9- Very broad, not clear, unspecific statement, doesn’t introduce the research topic. Please rephrase and add reference(s)
X10- Reference?
X11- Efficiency of what?
X12- What about the last decade? Since 2012 there is a ban on conventional cages, which led to more space per bird. Also, there is a worldwide shift towards cage free systems for laying hens. Hence, in many regions the useable surface per hen is actually increasing.
AP- We agree with the reviewer that starting lines are very broad and not clear. Therefore current text has been omitted and new introduction has been phrased.
X13- The ban on beak trimming is only very recent, whereas the feather pecking problem was already existent prior to the ban. Hence, feather pecking was/is also a problem in flocks where beak trimming is still allowed. The problem in non-beak trimmed flocks is bigger because the hens can peck more effectively.
AP- In order to shape the argument, ‘especially…………is forbidden’ is deleted from the text.
X14- Most likely found in more than 1 study.
AP- Another reference added to support this fact (Giersberg et al., 2017).
X15- This makes it more clear that feather pecking needs to be prevented at an early stage / young age!
AP- Already indicated this fact at line 50.
X16- (c) and (d) are introduced in the introduction, (a) and (b) are not. Add the possible interest (a) and (b) in relation to topics (feeding live BSF)
AP- Topics introduced in line 75 and 70.
X17- It is of interest to mention the previous housing system. If the difference of housing system is very big (e.g. previously housed in colony cages) this may have an effect on the behavior and performance.
AP- Ok. Included in line 87.
X18- How come? Any possible effects / correction on pecking observations?
AP- Birds were housed at the facility at 65 weeks of age with 22 birds per pen. Trials started at 67 weeks of age. In this 2 weeks of adaptation period, 1 bird died, making that at the start 21 birds in one pen. However, the effect of such event of production performance, egg quality, behaviour and feather pecking is expected to be minor. For the sake of uniformity, we have made 22 birds per pen.
X19- In results section it is stated that hens spent more time on the floor suggesting that time spent on the floor was observed, whereas here in the M&M is described the birds were counted (at 5 min. interval) which gives an output of number of birds on the floor. Number of birds on the floor is not the same as time spent on the floor.
AP- Agree with the reviewer. Modifications (line 253) have been done in results and discussion (line 341) sections to highlight this.
X20- Which Ethics Review Committee?
AP- Schothorst Feed Research Institute. Inserted in text.
X21- How relevant is the detailed information of the feed content in the Materials and Methods and in the Results if the outcomes are not used for interpreting he outcomes in the Discussion or Conclusions?
AP- We agree with the reviewer. Data used for generation of table 6 (previously table 8) is already available in table 3. Therefore section 3.1 is now removed from the MS.
X22- Give the P-value. Only if P<0.001 this can be written sown as such.
AP- Ok. Required changes are included.
X23- Add values ± sem
AP- Ok. Required changes are included.
X24- Figure can just as well be implemented in Table6 ± sem. If figure 1 is maintained then add error bars with sem
AP- We agree with reviewer. Required changes are included. Figure 1 is deleted and contents merged with table 5 (previously table 6).
X25- Give the P-value. Only if P<0.001 this can be written sown as such.
AP- Ok. Required changes are included.
X26- This is a strong trend and needs to be mentioned as such and discussed in the discussion.
AP- We agree with the reviewers. The trend in now discussed in discussion (line 271).
X27- Add ± sem per variable.
AP- Ok. Required changes are included.
X28- <Related to changes in table>
AP- Changes done.
X29- ± sem
AP- Now the figure is converted into table and SEM is added.
X30- Add ± sem per variable outcome.
AP- Ok. Required changes included.
X31- Very strong trend!! Must be in consideration when discussing the results.
AP- We agree with the reviewers. The trend in now discussed in discussion (line 308).
X32- Give the P-value. Only if P<0.001 this can be written sown as such.
AP- Ok. Required changes are included.
X33- Add values in the text between brackets and with ± sem.
AP- Ok. Figure 3 in converted into a table and sem is now included.
X34- ± sem.
AP- Ok. Included.
X35- Add error bars ± sem.
AP- See comment 33.
X36- …………spend more time on floor………….
AP- Changes done as discussed in comment 19.
X37- In line 96 it is stated that the larvae dispensed over a 6 hour time period. This does not match with this statement in line 254 and with the results found.
AP- We wanted to say ‘after loading of larvae in dispenser’. Changes done to highlight this.
X38- No new results should be presented in the discussion. Move these to the results sections.
AP- Ok. Table 8 is now moved to section 3.1 (production performance).
X39- This should be in the introduction and/or be mentioned here more briefly than how it is presented here. And/or could be moved to implications.
AP- We agree with the reviewers. In order to structure the MS better, we have briefed the argument at the current position itself.
X40- You don’t know if the egg quality parameters were unchanged, because then you should have compare those parameters with the parameters prior to the trial.
AP- We disagree with the reviewer. Egg quality was determined at the start and end of the trial, so it is a fair comparison for egg quality, and the statement that egg quality is not affected by the treatments is correct (in opinion of authors). We agree with the reviewer that it could have been better to analyse the egg quality before the starting of trials. However, it is very unlikely that the dietary treatments will have effect on egg quality within few days.
X41- Start with discussing the results that were found.
AP- Ok. Required changes are included.
X42- The difference of feather condition (strong trend) needs to be mentioned/discussed.
AP- Ok. Strong trend discussed in line 308.
X43- In the Netherlands and likely in other regions too it is quite common to keep brown layer hens to 80-85 week.
AP- Ok. Required changes done and reference update.
X44- What about offering enrichments, lowering stock density, free range, adapted feed formulations? And why do these interventions reduce the incidence of feather pecking?
AP- Ok. Offering enrichments, lowering stock density and adapted feed formulations have been included in the text. Free range systems are alternative systems and are discussed later in the same section.
X45- Should this section not be at the end of 4.4?
AP- OK. Required changes done.
X46- What about dispenser in line 94-95?
AP- We found that 6 h was usually effective. But maybe longer periods or at certain set points would be more beneficial. However, we have not included this briefing in text.
X47- (c). the additional costs involved in relation to the benefits?
AP- Ok. Included in the text. Frankly speaking our company already has a product called ‘OERei’ in market (available in Albert Heinj and Jumbo, Netherlands). The cost is little extra, but at the end customer is willing to pay it.
X48- Reconsider source
AP- Ok. New reference included.
X49- Be consistent in layout.
AP- OK.
Thanks and regards,
Aman Paul

Reviewer 2 Report
Overall a very interesting article, which takes up a current issue.
The paper is acceptable for publication with the suggested revision listed below.
The title and the abstract don't achieve the aim of your research (see Line 71-71), I suggest to reformulate it.
Materials and methods:
I suggest to combine the paragraph 2.1 and 2.5.
Line 150: Bird behavior: please add references.
Line 194-196: you should compare also the control treatment on dry matter basis
Line 263: Table 8. I suggest to put this table in the results.
Line 270-277: I suggest to emphasize this concept in the simple summary too
References:
Please check the references, there are many mistakes (missing pages, abbreviations incorrect..)
It would be interesting to insert a photo of the larvae dispenser.
Author Response
Dear Reviewer 2,
We really believe that your comments positively improved the quality of the manuscript. Your comments are visible in red text and yellow shading in attached manuscript (all markup version of word file). Below is the reply to all your comments:
The title and the abstract don't achieve the aim of your research (see Line 71-71), I suggest to reformulate it.
AP- Ok. The title has been reformulated.
Materials and methods:
I suggest to combine the paragraph 2.1 and 2.5.
AP- Ok. Paragraphs have been merged.
Line 150: Bird behavior: please add references.
AP- No specific reference or protocol. We only determined if birds were on floor or not every 5 mins.
Line 194-196: you should compare also the control treatment on dry matter basis
AP- Feed has approximately 110g moisture/kg. DM intake for groups A is 118.1 g/h/d and group B is 109.3 g/h/d. The authors do not want to show DM in the text because differences are relatively small.
Line 263: Table 8. I suggest to put this table in the results.
AP- Ok. Now Table 8 is moved to results.
Line 270-277: I suggest to emphasize this concept in the simple summary too
AP- Ok. Now also included in simple summary.
References:
Please check the references, there are many mistakes
(missing pages, abbreviations incorrect..)
AP- Ok. Everything will be thorough checked.
It would be interesting to insert a photo of the larvae dispenser.
AP- Ok. Photo included.
Thanks and regards,
Aman Paul
